# Introducing “MEW2” Software: A Tool to Analyze MQ-NMR Experiments for Elastomers

**DOI:** 10.3390/polym15204058

**Published:** 2023-10-11

**Authors:** Fernando M. Salamanca, Zenen Zepeda-Rodríguez, Laura Diñeiro, Marina M. Escrivá, Rebeca Herrero, Rodrigo Navarro, Juan L. Valentín

**Affiliations:** Instituto de Ciencia y Tecnología de Polímeros (ICTP-CSIC), Juan de la Cierva 3, 28006 Madrid, Spain; zenen@ictp.csic.es (Z.Z.-R.); ldineiro@ictp.csic.es (L.D.); mmonteroescriva@gmail.com (M.M.E.); rherrero@ictp.csic.es (R.H.); rnavarro@ictp.csic.es (R.N.); jlvalentin@ictp.csic.es (J.L.V.)

**Keywords:** rubber network, software, automatization, low-field NMR spectroscopy, MQ-NMR, vulcanization

## Abstract

Low-field time-domain proton Nuclear Magnetic Resonance (NMR) spectroscopy is an attractive and powerful tool for studying the structure and dynamics of elastomers. The existence of crosslinks and other topological constraints in rubber matrices (entanglements and filler–rubber interactions, among others) renders the fast segmental fluctuations of the polymeric chains non-isotropic, obtaining nonzero residual dipolar couplings, which is the main observable of MQ-NMR experiments. A new software, Multiple quantum nuclear magnetic resonance analyzer for Elastomeric Networks v2 (MEW2), provides a new tool to facilitate the study of the molecular structure of elastomeric materials. This program quantitatively analyzes two different sets of experimental data obtained in the same experiment, which are dominated by multiple-quantum coherence and polymer dynamics. The proper quantification of non-coupled network defects (dangling chain ends, loops, etc.) allows the analyzer to normalize the multiple quantum intensity, obtaining a build-up curve that contains the structural information without any influence from the rubber dynamics. Finally, it provides the spatial distribution of crosslinks using a fast Tikhonov regularization process based on a statistical criterion. As a general trend, this study provides an automatic solution to a tedious procedure of analysis, demonstrating a new tool that accelerates the calculations of network structure using ^1^H MQ-NMR low-field time-domain experiments for elastomeric compounds.

## 1. Introduction

Elastomers are unique polymeric materials characterized by a long-range elasticity obtained after a process called vulcanization [1,2,3]. As the crosslinks formed during this process are permanent, it is able to generate materials with high technological interest from the point of view of industry. Theredore, the properties and performance of elastomers are directly related to the polymer structure in terms of the number and nature of crosslinks, filler–rubber interactions, the spatial distribution of constraints, and the fraction of network defects (mainly dangling chain ends and loops). Characterization of the network structure of those materials can be carried out from several different experimental approaches, the most well known of which are swelling experiments and dynamic mechanical properties. However, time-domain solid-state MQ-NMR spectroscopy has been demonstrated over the years [4,5] to be a versatile, powerful, and successful tool for investigating the network structures of different rubber materials. MQ-NMR experiments have proven to be a good quantitative experimental approach for obtaining the network parameters (including the crosslink density) needed to understand the macroscopic behaviour of different rubber compounds. In combination with other experimental techniques, they represent a powerful tool that allows researchers to analyze the unique properties that elastomers exhibit on the macroscopic scale from a molecular level point of view [4]. The theoretical background of MQ-NMR experiments has been widely explained in previous works [5,6,7]. Several works [8,9] have been published that use general tools for analysis of the data with tools that are not optimized for MQ-NMR experimental data, such as Origin^®^. While these calculations remain undisputed, in this work we present a new tool that automates the process of analysis. Our main objective is to present a new program that can save time and avoid human error. This software is very useful and suitable for analyzing the data from MQ-NMR experiments in an almost automatic way.

The importance and application of MQ-NMR approaches to obtain the macromolecular behaviour of different compounds are based on the capability of MQ-NMR to elucidate the molecular structure while distinguishing between structure and dynamics. A large number of advances have been developed using this technique in the study of crosslinked networks with different matrices, crosslinking systems, and additives. In addition, the role of entanglements in network properties has been discussed [10,11,12,13,14]. Other examples of the versatility of MQ-NMR experiment include its use in unveiling molecular details across a range of polymeric materials [15,16], such as swollen gels [17,18], nanocomposites [19], polymer melts [20], and rubber blends [21]. Finally, it should be noted that the unique quantitative information from MQ-NMR experiments has been crucial in determining the degree of devulcanization in several types of rubber, leading to enhanced molecular characterization of recycled rubbers through this powerful technique [22,23,24].

The main observable that can be obtained from the analysis of MQ-NMR experiments is the residual dipolar coupling ($D_{res}$), which is a quantity that results as a time average over the fluctuations of the dipolar tensor covering the time until the plateau in the correlation function is reached. This contribution behaves as a constant for a certain temperature and timescale when a macromolecular network structure is held. The characteristics of the plateau are provided by the square of the order parameter of the polymer backbone:(1)$$S_{b}=\frac{D_{res}}{D_{stat}k^{-1}}=\alpha \frac{r^{2}}{N}$$
where $S_{b}$ is the order parameter, $D_{res}$ is the residual dipolar coupling, $D_{stat}k^{-1}$ is the static limit dipolar coupling constant based on a parameter *k* that contains information about the intra-segmental motions, α is a constant that depends on the assumptions of the distribution of the mobile chains (for Gaussian statistics, α=0.6), $r^{2}$ is the ratio of the end-to-end vector of the chain segment with respect to the non-deformed melt state (therefore, $r^{2}=1$ for non-stretched samples), and *N* is the number of Kuhn segments between two consecutive constraints [25].

It can be seen from Equation (Equation 1) that while $r^{2}$ remains constant for the same elastomeric matrix (as well as for the same elongation and swelling rate), variations in $D_{res}$ are directly related with *N*. This means that $D_{res}$ can be directly related to all the topological density constraints (including both crosslinks and entanglements) for a certain microscopic model. However, the most accepted models are the affine model, the phantom model [3], and more recently the Lang–Sommer model [26,27]. All of these assume that *N* can be related to $N_{c}$ (the number of Kuhn segments between two consecutive crosslinks) while the contribution of the entanglements remains constant. At this point, the assumption of the entanglement contribution only depending on the elastomeric matrix and its molecular weight is accepted, although it remains an open issue [28] that exceeds the objectives of this work.

In this work, we present the basic aspects of MQ-NMR experiments analysis, introducing the details in the Theoretical Background section, followed by an extended explanation of the performance of the program applied to real experiments. Subsequently, we compare the results of this program and the standard methods, and conclude that software described in this paper can provide the same network structural parameters at a much lower time cost.

## 2. Theoretical Background

MQ-NMR is a well-established experimental approach that provides molecular-scale information on the degree of cross-linking of an elastomer (more quantitative than the traditional Hahn-echo $T_{2}$ relaxometry) [6]. The main observable is the so-called residual dipolar coupling ($D_{res}$), which is a fast-limit dynamic average of the instantaneous and orientation dipole–dipole coupling between the protons fixed to the polymer backbone and is measured in the elastomeric state far above the glass transition [29]. It quantifies the degree of anisotropy of the segmental movement, which is model-dependent with the inverse number of segments between crosslink or entanglements constraints (see Equation (Equation 1)). The simplest models consider an inversely proportional relationship between $D_{res}$ and $N_{c}$ (or the associated molecular weight $M_{c}$) and a proportionality constant (see Equation (Equation 2)) [6]:(2)$$M_{c}^{NR}\left(\mathrm{kg}\;{\mathrm{mol}}^{-1}\right)=\frac{617}{D_{res}\left(\mathrm{Hz}\right)}$$
where $M_{c}^{NR}$ represents the molecular weight between two consecutive crosslinks and $D_{res}$ is the residual dipolar coupling in Hz.

The relative width of the distribution of $D_{res}$ and the fraction of non-elastically active material are proofs of the technique’s sensitivity to the degree of local inhomogeneity. As a result of the MQ-NMR experiments, two distinct sets of data can be acquired: a double quantum (DQ) build-up curve $I_{DQ}\left(\tau_{DQ}\right)$, and a reference intensity decay curve $I_{ref}\left(\tau_{DQ}\right)$. An MQ-NMR experiment involves the application of the DQ Hamiltonian, which usually assumes the excitation of all even quantum states orders, hence the terminology MQ-NMR. At short pulse sequence times, the initial growth of the build-up function is dominated only by double quantum coherence (DQ), while at longer evolution times it is determined by higher 4n+2 quantum orders as well (n≥0). On the other hand, the reference curve’s decay is determined by the quantum order n=0, which represents the dipolar-modulated longitudinal magnetization and contains information from both superior quantum orders (4n) and the non-elastically active fraction of the sample. Additionally, the NMR signals $I_{DQ}$ and $I_{ref}$ are affected by the polymer dynamics.

With the two signal functions, one can extract suitable information of the polymer network independently of temperature related relaxation effects (polymer dynamics) by the application of a normalization process to the MQ build-up curve. The latter requires the proper identification, quantification, and subtraction of low-decaying signal tails deriving from the non-coupled protons (e.g., the fraction of network defects). As a result of the equal partitioning among all excited even quantum orders in the large evolution time regime, the normalized MQ build-up curve reaches a plateau of 0.5 in the large evolution time regime, which is an independent approximation regardless of temperature. The residual dipolar coupling distribution (which includes the structural information for the polymer network) is directly derived by a fast Tikhonov regularization process.

Until the development of the MEW2 software, the analysis of the data extracted from MQ-NMR experiments was performed using other software, such as Origin^®^, Microsoft Excel^®^, or Scidavis^®^, the last being a free application developed for scientific calculations. As the protocol used for analysis is very similar among all these options, it is briefly described here to provide a better understanding of the human mistakes that can be avoided with automated software and the time savings available with the new software that we present here.

First, the user must load the MQ-NMR data file to their preferred software. The user has to identify the columns as follows: the first column contains the DQ evolution time, usually measured in ms ($\tau_{DQ}$), the second column can be identified as the reference intensity $I_{ref}$, and the third column contains the double quantum intensity $I_{DQ}$.

Then, the re-scaling of the intensities and the tail subtraction have to be performed by dividing both intensities by the first value of $I_{ref}$. The tail subtraction is obtained from a semi-logarithmic plot of the subtraction intensity $I_{sub}$, which can be calculated as follows:(3)$$I_{sub}=I_{ref}-I_{DQ}$$
where $I_{ref}$ represents the reference intensity, $I_{DQ}$ represents the double quantum intensity, and $I_{sub}$ is the subtracted intensity obtained by the subtraction of the first ones.

While the mathematical form of Equation (Equation 3) is complex due to the different effects of dynamics and structure at the molecular level, it can be assumed that the first fast decay in the intensity comes from the rubber network, which is usually the most prominent because of the dipolar coupling effect. On the other hand, the second decay in the subtracted intensity has a lower slope (slower decay) and is correlated with the dynamics of the non-coupled network defects, or more generally the fraction of the sample containing dangling chain ends and small molecules (e.g., processing oils or plasticizers). This component is called $I_{tail}$. Due to the differences between the molecular structure and composition of the wide range of rubber compounds that can be analyzed by MQ-NMR, $I_{tail}$ can show different shapes, as is discussed in the next section, although the most usual case for rubber samples is to observe an exponential behaviour:(4)$$I_{tail}=A\cdot e^{-\tau_{DQ}/\tau }$$

The user has to manually select the optimum evolution time interval at which the non-elastically active fraction of material dominates, then perform a nonlinear fitting using the assumed model. Afterwards, it is necessary to calculate the sum intensity mentioned above ($I_{\Sigma MQ}=I_{DQ}+I_{ref}$), which includes all the excited quantum orders encoded in $I_{ref}$ and $I_{DQ}$ in half. Nevertheless, it is required to subtract the intensity from the non-coupled protons ($I_{tail}$). In this way, it is possible to perform a point-by-point normalization for the DQ intensity that only contains structural information, that is, the normalized DQ intensity ($I_{nDQ}$):(5)$$I_{nDQ}=\frac{I_{DQ}}{I_{\Sigma MQ}-I_{tail}}$$

When $I_{nDQ}$ has been calculated, the user builds the file ftikreg.dat, which is a main input for ftikreg_2.00.exe [5]. One of the fundamental motivations of using a regularization process and not a nonlinear fitting is that it is impossible to accurately fit a function for inhomogeneous polymer networks with considerable relative width or even multimodal distributions of dipolar couplings. For example, this may occur in swollen polymer networks, network chains with spatially distant bimodal or multimodal chain length distributions, and filled samples with high filler–rubber interaction contributions. This is because only a unique residual dipolar coupling constant is considered. A Fredholm integral equation usually provides a better approach:(6)$$g\left(\tau_{DQ}\right)=\int_{0}^{\infty }K\left[D_{res},\tau_{DQ}\right]f\left(D_{res}\right)dD_{res}$$
where $g(\tau_{DQ})$ represents the measured data, $f(D_{res})$ is the residual dipolar coupling distribution, and $K[D_{res},\tau_{DQ}]$ is the kernel function, which is approximated from [5]
(7)$$I_{nDQ}\left(\tau_{DQ}\right)=0.5\left[1-e^{-{\left(0.378D_{res}\tau_{DQ}\right)}^{3/2}}\cdot \cos\left(0.583D_{res}\tau_{DQ}\right)\right]$$

Then, although *f* distributions are not directly accessible via experimental data, they are often connected to a measurable quantity *g* by an operator equation:(8)g=Af.

Therefore, if $A^{-1}$ exists, then
(9)$$A^{-1}g=f.$$

If $A^{-1}$ is discontinuous, then Equation (Equation 9) is called ill-posed and the solution is only available via numerical approaches. A variation algorithm proposed by Tikhonov quantifies the regularization parameter α, which contains information about the smoothing and shape of the obtained distribution and stabilizes the solutions. In addition, it has been found that
(10)$$\det(Af_{\alpha }-g)=\epsilon ,$$
where ϵ is the noise level. One of the advantages of Tikhonov regularization is that it calculates the parameter α; therefore, the distribution of residual couplings for a given error ϵ that results from the calculation is associated with that α value.

For a reliable distribution of coupling constants, a robust criterion needs to be determined, which should be independent of the error parameter applied in the regularization process.

In this case, the approximate nature of the improved kernel function arises from minor systematic errors, adding to the problem of the error parameter dependence of the regularization process. For this reason, an alternative protocol is suggested and implemented in a convenient way via new subroutines with the sole purpose of making the process easier to the final user.

A first step in the process consists of automating the error parameter ϵ. Using logarithmic equidistant steps, the regularization can be performed in successive calculations within a given error interval range. Each distribution $f(D_{res})$ is converted to a build-up curve $g(\tau_{DQ})$, assuming the same kernel function as for regularizing the inverse problem by directly evaluating Equation (Equation 6) in a discrete form. A $\chi^{2}$ test calculates the mean square deviation between the fit and data, and is needed to compare the experimental build-up curve obtained from the distribution resulting from the regularization process to quantify the precision of the distribution. Then, the final results are the spatial distribution of the constraints $D_{res}$ and the build-up curve $I_{nDQ}$, which should be similar to the one from the experimental data. This automated process is carried out using the subroutine compiled in ftikreg, previously published in [5] and adapted from [30].

Several files are provided when the regularization protocol finishes; while in the previous standard method the user had to handle them manually, plot them, and extract all the valuable information, the new MEW2 tool is does this automatically, saving time and avoiding systematic errors due to human factors. This feature becomes very useful when the user needs to analyze a considerable amount of samples in a short periods of time.

## 3. Algorithm Performance

One of the main aspects of Multiple quantum nuclear magnetic resonance analyzer for Elastomeric netWorks v2 (MEW2) is that it works under any Windows version thanks to the compatibility of the builder from the Python script to the executable application. The executable includes all the dependencies that are needed for the correct development of the analysis of the data provided by the MQ-NMR experiment, excluding the regularization process. Thus, the files ftikreg.dat, ftikreg.par, and ftikreg_2.00.exe are needed, and must be located in the same folder as both the application and the .txt file from the MQ-NMR experiment.

The file that contains the MQ-NMR data should be a plain-text file ordered by columns and with spaces as separators, where the first column is the evolution time in ms ($\tau_{DQ}$), the second column is identified as the reference intensity $I_{ref}$, and the third column is the DQ intensity, called $I_{DQ}$. The lengths of the columns must be the same; this specification is controlled from the low-field spectrometer software.

The first feedback that the user receives after launching the software is the raw data provided by the NMR experiment. Figure 1a shows the reference and multiple-quantum intensities as a function of $\tau_{DQ}$. The next step is to re-scale both intensities to the first value of $I_{ref}$. The normalisation leads to a re-scaling of the experimental data; therefore, the shape of both figures remains unchanged, as shown in Figure 1b. Both figures are direct output from the MEW2 software.

As shown in Section 2 (Theoretical Background), it can be assumed that the description of the investigated material consisting only of elastically active network chains is inaccurate. Usually, additional components such as short dangling chain ends, sol fraction, oils, and other small molecules may be present. The dynamics of these components are fast enough to exhibit an isotropic movement in the NMR timescale, providing zero residual dipolar coupling. These contributions are only detected in the reference data, and as such form more slowly decaying long-time tails. In order to successfully separate the dynamic and the structural parts while observing the 0.5 plateau for $I_{nDQ}$, it is crucial to remove these contributions through suitable fitting while attending to a mathematical model, followed by subtraction of the tails [6].

Figure 2 shows a semi-logarithmic plot of $I_{sub}$ against $\tau_{DQ}$, where a fast-relaxing phase corresponding to the network contribution and a slower-relaxing phase corresponding to the amount of network defects, sol fraction, solvents, etc., can be observed. Both figures are plotted and displayed by the MEW2 software, and are stored in a folder when the analysis is finished. It is important to note that the tail subtraction is independent of the visualization scale of the plot.

Therefore, the fraction of these contributions within the analysed material is encompassed under the same exponential fitting. However, owing to the high versatility of MEW2, the software provides three different fitting modes for this step; in addition to the model, the user can choose the time interval (see Figure 3) in which the isotropic dynamics regime dominates. The three fitting modes are as follows:A single exponential fitting:
(11)$$I_{\mathrm{tail}}=A\cdot e^{-\tau_{DQ}/\tau }$$A stretched exponential with stretched coefficient *b*
(12)$$I_{\mathrm{tail}}=A\cdot e^{-{\left(\tau_{DQ}/\tau \right)}^{b}}$$A two-component exponential fitting
(13)$$I_{\mathrm{tail}}=A_{1}\cdot e^{-\tau_{DQ}/\tau_{1}}+A_{2}\cdot e^{-\tau_{DQ}/\tau_{2}}$$

**Figure 3 polymers-15-04058-f003:**
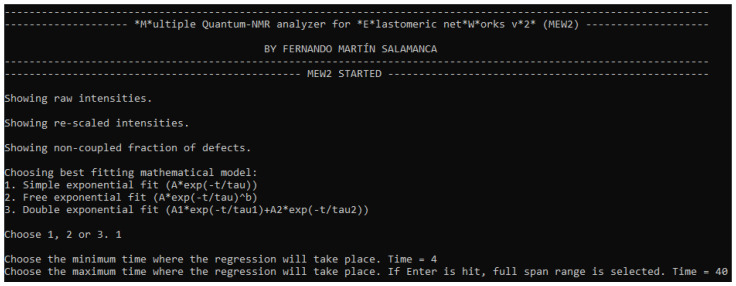
Screenshot of the output, where the user can choose the fitting model and the interval for the $T_{2}$ isotropic component.

The selection of the tail model depends on the material or, in other words, on the molecular structure and composition of the non-coupled fraction. Highly plasticised compounds might need a complex model (Equation (Equation 13) [31]), while most of the vulcanized compounds analyzed by MQ-NMR in the literature [22] only exhibit one component decay, as shown in Figure 2. Figure 2 shows that a constant behaviour in the long time regime contribution might be ignorable due to the low contribution to the $I_{nDQ}$ signal (lower than 1%). This means that the optimum model is the single exponential shown in Equation (Equation 11) [31] for most cases.

After the mathematical model for the tail has been chosen, the use is asked for the range in $\tau_{DQ}$ and introduces the desired limits for the fitting, leading to the red fitting line in Figure 2, where *A* is the fraction of non-coupled network defects, τ is the relaxation time for the slow-relaxing phase, and *b* is the independent exponent. The expression chosen for the fitting is shown in the picture, as the fitting shown by the a red line is only a visual guide indicating the quality of the fitting. It is important to note that if no end-point is selected for the interval of the tail subtraction, all the registered points are used. Then, the normalization process is applied to obtain the build-up curve from the experimental data:(14)$$I_{nDQ}=\frac{I_{DQ}}{I_{ref}+I_{DQ}-I_{\mathrm{tail}}}$$

Figure 4 shows $I_{nDQ}$ as a function of $\tau_{DQ}$, obtaining the plateau in 0.5, which means that the structure and the dynamics have been successfully separated. If the plateau is not obtained, then the tail subtraction must be revised [6]. Then, the data of $I_{nDQ}$ and $\tau_{DQ}$ are collected in the ftikreg.dat file and the regularization process starts, as described in [5]. The non-integration of the Tikhonov regularization files in the MEW2 application have the advantage of allowing the user to modify ftikreg.par if needed. As it is the configuration file for the ftikreg_2.00.exe application, it can be modified as required, for example to customize the kernel function or (depending on the material) the minimum or maximum dipolar coupling limits. A more detailed information about all the features that ftikreg_2.00.exe provides can be found in the Supporting Information of [5].

After the preparation of the ftikreg.dat file that respectively contains the evolution time in ms and $I_{nDQ}$ in the first two columns, the ftikreg.par file is modified to optimize the parameters of the regularization process. The regularization process then takes place and the obtained results for the $\chi^{2}$ statistics are plotted and displayed (see Figure 5a). Next, the console requires the interval of the calculated distributions that the user wants to display onscreen. It should be taken into account here that if the chosen distribution is typically on the low-$\chi^{2}$ region it may be affected by mathematical artifacts or results that might not have any physical meaning, as already explained in the previous sections and in [5].

Figure 4 and Figure 5a–c report the plots shown to the user during the analysis process; if the user decides to save the analysis data, all the plots are stored in a folder with the name of the MQ-NMR file. Alternatively, it is possible to save each of them individually when they are displayed. The distributions are shown in the form of the example in Figure 5b, where $\bar{D_{res}}$ is the average value of the distribution, $D_{res}^{*}$ is the maximum value of the distribution, and δ is the relative width, calculated as
(15)$$\delta =\frac{\sigma }{\bar{D_{res}}}$$
where σ is the standard deviation of the distribution. The fraction of previously calculated defects is shown as well.

Finally, the user can choose the distribution that best suits the physical behaviour of the material; the information of this distribution is stored in a folder along the same route as the files (see Figure 6). In addition, a comparison between the experimental data and $I_{nDQ}$ as determined by the regularization process is shown (see Figure 5c) in order to compare the agreement between the mathematical method and the experimental data. The consistency of the comparison between the experimental $I_{nDQ}$ and the one provided by the regularization process should be considered in the region where $I_{nDQ}<0.45$, which is the region that mostly determines the calculation of $D_{res}$. A maximum in $I_{nDQ}$ can be observed for narrow distributions, and does not have a considerable impact on the final results [6].

All the information obtained during the analysis is stored in a folder if the user agrees (the console asks before saving), making it possible to work with the obtained data after the analysis is finished. The amount of time spent using MEW2 is less than calculations carried out “by hand”, while the results do not seem affected. Moreover, the user can actively interact with the software at each step to guarantee control over all of the steps in the analysis.

## 4. Experimental Section

The sample called “Rubber” consists of standarized poly(cis-1,4-butadiene) rubber (BR) with 98% cis-1,4 structures, and was obtained from Polimeri Europa. This sample was cured using a conventional vulcanization system based on a sulfur–accelerator recipe containing zinc oxide (5 parts per hundred (phr) of rubber) and stearic acid (2 phr) as activators, N-cyclohexyl-2-benzothiazolesulphenamide (CBS) as accelerator, and sulfur (7 phr). The sample was prepared in an open two-roll mill using standard mixing procedures and vulcanized in a laboratory press at 160 ºC at the respective optimal times ($t_{97}$) deduced from the rheometer curve (Monsanto moving die rheometer, model MDR 2000E).

The sample of filled rubber was kindly supplied by Birla Carbon (part of the Aditya Birla Group); it consists of a natural rubber (NR) reinforced with 50 phr of carbon black of grade N330 and vulcanized with 2.5 phr of sulphur as part of a conventional vulcanization system.

The sample of polyurethane (PU) was kindly supplied by Valora Teruel S.L. (Zaragoza, Spain); this is a commercial product called Maflex 15. The compound is an elastic polyurethane highly plastified with chlorinated paraffins and intermediate molecular weight. The polymer is obtained by mixing similar volumes of a prepolymer based on TDI and a polyfunctional polyol based on propylene oxide.

The sample of end-of-life tire powder (ELTp) was kindly supplied by Valoriza Medioambiente (Grupo Sacyr S.A. Madrid, Spain). The powder presented a nominal particle size of 550 µm and was composed of different truck tire parts (e.g., inner liner, tread, sidewalls, etc.) containing different chemical formulations.

All the MQ-NMR experiments were carried out on a Bruker minispec mq 20 (Larmor frequency 20 MHz, $B_{0}=0.5\;T$) at 80 °C with a 90° pulse length of 2.2 µs and a dead time of 13 µs. The experiments and the analysis of the measured raw data were performed following the previously published procedures [22,32,33], including subtraction of the small contributions of signal tails related to network defects prior to calculating $I_{nDQ}$.

## 5. Results and Discussion

To demonstrate the versatility and potential of the new MEW2 tool, elastomeric samples with different natures were analysed, from vulcanised rubber samples to highly plasticised polyurethanes. All the samples described in the previous sections were analysed in a low-field NMR spectrometer. For comparison, the structural and dynamic information of these polymeric networks was extracted from the signal sets using the long and tedious Origin ^®^ protocol described above as well as using the MEW2 software. The results of the analysis are reported in Table 1.

The results show an agreement of more than 95% between the two methods in therms of relative errors, while MEW2 takes less time to achieve the same results. The measurements of relative width should consider the propagation of the errors derived from Equation (Equation 15), where it is clear that the obtained results are basically the same for the same dataset.

Figure 7 shows the agreement between the two methods and the versatility over a wide range of residual dipolar coupling (1000 Hz for the studied samples) that can be obtained with MQ-NMR experiments applied to elastomers. The differences in width, skew, and other quantities related to the distributions can be related to the molecular structure of each material.

## 6. Conclusions

We have developed a semi-automatic tool to analyze the output data from MQ-NMR experiments. This attractive tool provides valuable information on the analyzed polymer network in a shorter time than other standard methods without affecting the final results. An experienced user can perform three analyses per minute using the new MEW2 software, while with the standard method described in Section 2 (Theoretical Background) each analysis takes much more time for the same type of user. It is important to note that nearly all of this time consumption take place while the regularization process is running. For this reason, an additional feature for the software in future versions might be the integration of the regularization process within the font code to avoid extra processes. It is important note that while inexperienced users may find it easier to work with an automated tool, their lack of experience might lead to wrong results; thus, training is highly recommended. Even with an automated tool, MQ-NMR analysis is a complex issue.

Another improvement for the future would be the addition of more than one way to obtain $D_{res}$, for instance, by fitting an Abragam-like function or through simultaneous fitting of two sets of data, as has been done in other works [34]. Further future developments of MEW2 could include different options for the analysis of various pulse sequences applied to elastomeric materials. These features could make this program a standard in the MQ-NMR analysis, where for now its main purpose is saving as much time as possible while providing a simple working version of the program which that can be improved in the future.

## Figures and Tables

**Figure 1 polymers-15-04058-f001:**
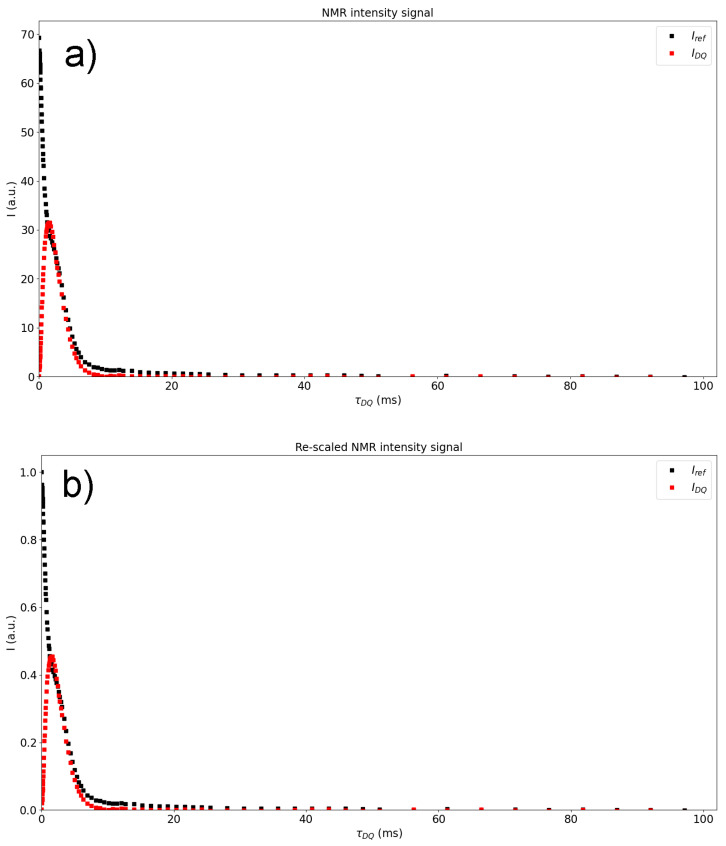
Reference and DQ intensities for a sample of Butadiene Rubber (BR) vulcanized with sulphur: (**a**) raw data obtained from the spectrometer and (**b**) data re-scaled to the maximum value of $I_{ref}$.

**Figure 2 polymers-15-04058-f002:**
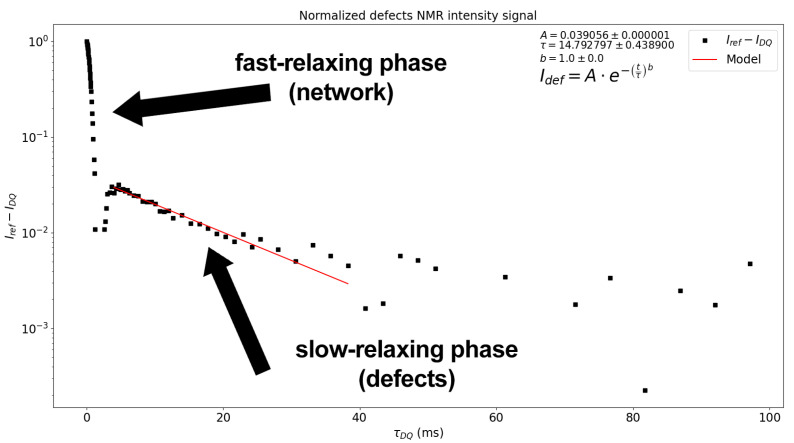
$I_{sub}$ plot with the best-fit exponential function of the tails (Model 1).

**Figure 4 polymers-15-04058-f004:**
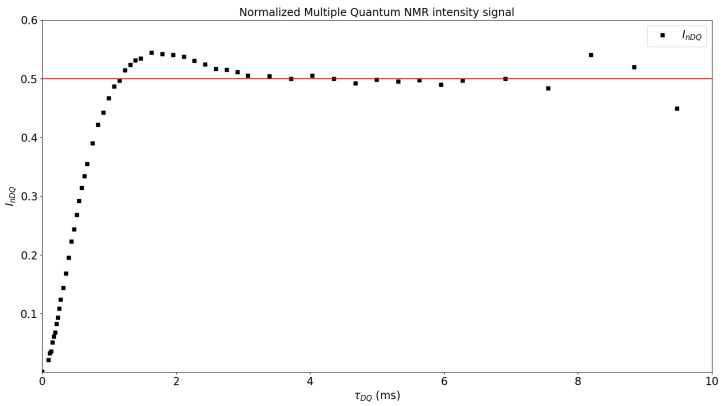
Normalized build-up curve; the red line is a visual guide.

**Figure 5 polymers-15-04058-f005:**
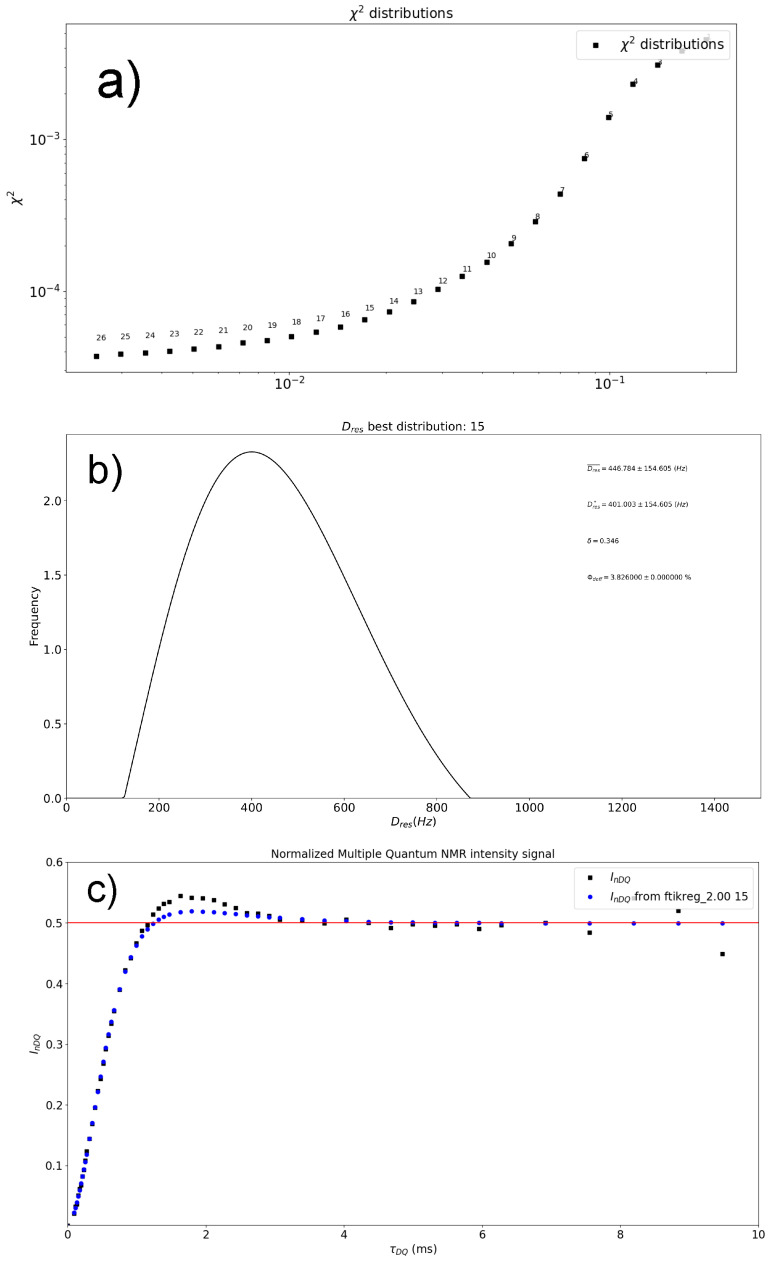
Regularization results: (**a**) $\chi^{2}$ values as a function of the error, (**b**) spatial distribution of $D_{res}$, and (**c**) $I_{nDQ}$ compared to the experimental data.

**Figure 6 polymers-15-04058-f006:**
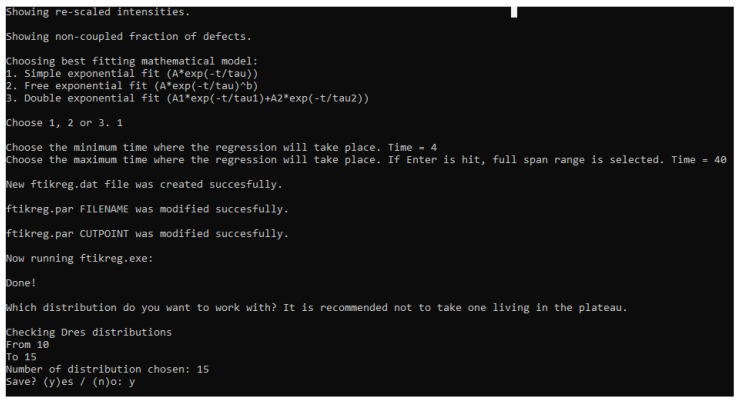
Output of the post-regularization steps.

**Figure 7 polymers-15-04058-f007:**
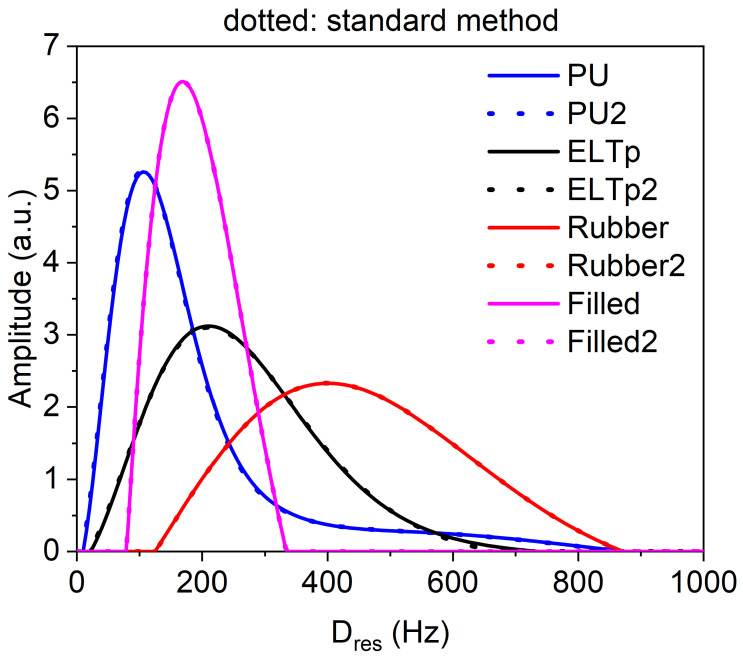
Spatial distribution of constraints among different elastomeric materials. The dotted lines were obtained with the standard method and the solid lines were obtained with the new MEW2 tool.

**Table 1 polymers-15-04058-t001:** Comparison between MEW2 analysis and standard analysis using Origin.

	Rubber		Filled Rubber		ELTp		PU	
	**MEW2**	**Std**	**MEW2**	**Std**	**MEW2**	**Std**	**MEW2**	**Std**
$D_{res}^{*}$ (Hz)	401	401	169	170	211	210	105	104
$\bar{D_{res}}$ (Hz)	447	446	189	189	264	262	189	187
δ	0.346	0.346	0.288	0.288	0.488	0.468	0.791	0.795
*A* (%)	3.82	3.58	3.62	3.94	14.86	14.76	57.33	59.48

## Data Availability

The data presented in this study are available on request from the corresponding author and Appendix A.

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
