# Peer review of "Introducing “MEW2” Software: A Tool to Analyze MQ-NMR Experiments for Elastomers"

_polymers, 2023, doi:10.3390/polym15204058_

Round 1
Reviewer 1 Report (Previous Reviewer 1)
I am satisfied with the revisions
Author Response
The authors thank Reviewer 1.
Reviewer 2 Report (Previous Reviewer 2)

Minor editing of English language required
Author Response
The authors thank Reviewer 2 for all the valuable comments and contributions. Please see the attachment.

Reviewer 3 Report (New Reviewer)
This manuscript presents a new tool to accelerate the computation of elastomer compound network structure using 1H MQ-NMR low-field time domain experiments, providing an automated solution for tedious analytical processes. The manuscript is clearly structured and logically flowing, but there are still some issues that need to be revised before it can be officially published.
1. It is recommended that the author adjust the line thickness of the images and the font size of the numbers in the manuscript.
2. Line 138, on the method of rescaling intensity, suggests that the author give an example, or briefly explain it with a formula, so that it is easy to understand.
3. It is recommended that the author add an explanation of the meaning of the letters in the formula below the formula.
4.Line 244, mentions that the fast and slow relaxation phases can be observed from Figure 2, and it is suggested that the author indicate the location of the fast and slow relaxation phases in the figure.
5. In lines 356 the manuscript, it is mentioned that both methods have a consistency rate above 95%, with MEW2 method achieving similar results in less time. It is suggested that the author provide an explanation combining relevant numerical values and results facilitate better understanding for readers.
6.The format of the table needs to be modified.
7. It is recommended to add the residuals of the upper fitted curve to Figure 2.
8. There are a lot of annotations and highlights in the article, pay attention to the format of the article.
9. It is suggested to show whether the method is universal and applicable to all elastomer network structures.

Author Response
The authors thank Reviewer 3 for all the valuable comments and contributions. Please see the attachment.

This manuscript is a resubmission of an earlier submission. The following is a list of the peer review reports and author responses from that submission.
Round 1
Reviewer 1 Report
This article reports a new software Multiple quantum nuclear magnetic resonance analyzer for Elastomeric Networks v2 (MEW2), which provides a new tool to facilitate the study of the molecular structure of elastomeric materials. This tool provides valuable information of rubber’s structure in shorter time, and greatly reduces the difficulty of obtaining information, which is enough to support its publication. However, there are several issues as below that the authors need to clarify, and publishing this manuscript in POLYMERS is reconsidered after the following minor revisions:
1. Can the reported software process the MQ-NMR data of filled rubber? If possible, please add relevant experimental evidence.
2. The highlight statement of the software is not outstanding enough.
3. Figure 5: It can be seen from Fig.5c that the consistency between the fitted curve and the original curve is not very ideal. Please explain whether this situation has any impact on the final result.
Reviewer 2 Report
The presented manuscript introduces a novel software, MEW2, that can be used to process data obtained from proton multiple-quantum (MQ) NMR experiments on elastomeric materials, that can be analyzed to extract residual dipolar couplings (Dres), an NMR observable related to the amount of cross-links in the material. Basically, the software performs the processing of MQ data to obtain normalized intensities of the double quantum intensities (InDQ), taking also into account the presence of defects. The software is then used to launch the ftikreg software, originally introduced by Weese et al. in 1992 and then modified by Chassé et al. in 2011, which performs the actual data analysis, by applying a Tikhonov regularization to the InDQ data, in order to obtain a distribution of Dres for the analyzed material.
General comments
The idea of introducing a software for a systematic processing of MQ-NMR data could be an interesting way to speed up their analysis (which is actually performed by the ftikreg software). However, I believe that the proposed work cannot be accepted for publication in Polymers, because of the following concerns:
1) More than 90% of the theoretical background section is copied from Chassé et al., 2011.
2) The authors do not give credits to the ftikreg software, which is the software launched by MEW2, which performs the actual analysis of the processed MQ-NMR to obtain Dres distributions.
Additionally, the introduction and results and discussion sections are really poor; the “Origin ® protocol” for the processing of the MQ-NMR data, which is used for comparison with the MEW2 software, is not described; finally, several English (and also formal) errors can be found throughout the text, which prevent the complete comprehension of the paper and make it really hard to read.
Notwithstanding my previous considerations, I hope the authors will take advantage of the following specific comments to rewrite a new manuscript on this topic.
Specific comments
1) The introduction is not clear and very poor. Since the proposed software is used to process MQ-NMR data, this experiment must be properly introduced also explaining what Dres are and their relevance in polymer science.
2) As mentioned in the general comments, it is unacceptable that the theoretical background section is copied from another paper.
3) Algorithm performance section: the authors must clearly claim that MEW2 consists in an executable that prepares the ftikreg.dat file for further analysis that are done by the ftikreg software. Appropriate credits to ftikreg must be added, with all the necessary references.
4) Line 104: which is the format needed for the .txt input file of the MQ-NMR data (e.g. first column: DQ evolution time expressed in ms; second column: IDQ; third column: Iref)? Report an example.
5) Figure 2 (b) shows the same plot of Figure 2 (a) with the addition of the fitting function of the tail. The caption describes something different.
6) Where do equations (5), (6) and (7) come from? Please, cite appropriate references.
7) Line 132: how do one choose between equations (5), (6) and (7) for tail subtraction? Are they dependent on the type of polymer used in the elastomer, additives, etc.? Please, discuss this.
8) Figure 3 (and 6): In the screenshot, it is reported “Choose the minimum time where the regression will take place” and then “Choose the last point where the regression will take place”. Please, be consistent and use either time (in this case, possibly indicate the unit to be used) or point.
9) After equation (8) (line numbers are missing): this part must be better explained and integrated with the citation of Chassé et al. 2011.
10) After Figure 4 (line numbers are missing): the text is barely comprehensible. Are figures reported in Figure 5 an output of the MEW2 software? Please, be more explicit and specific.
11) Line 178: which “previously cited procedures”? Add the number of the references.
12) Line 186: the description of the “long and tedious Origin ® protocol” must be added for an appropriate comparison. Moreover, for each of the two protocols used to process the data, the quantification of the time and effort needed for the user should also be reported.
13) Prior to the comparison of the output obtained from the ftikreg analysis, It would be better to compare the InDQ curves obtained using the two different approaches. Please, add this figure.
14) It would be nice to add the analysis of other types of elastomeric materials, also including the most “tricky” ones from the point of view of MQ-NMR data processing, such as lowly crosslinked elastomers based on styrene-butadiene rubbers.
15) The conclusions must be completely revised according to all previous comments.
16) Lines 204-205: “It could be possible to add more than one way to obtain Dres”. Is it possible right now or not?
17) Lines 206-207: The presentation of an implemented software with these functions would certainly add more value to a new rewritten paper.
Several English (and also formal) errors can be found throughout the text, which prevent the complete comprehension of the paper and make it really hard to read.